# Body Composition and Biological Functioning in Polish Perimenopausal Women with Type 2 Diabetes

**DOI:** 10.3390/ijerph182111422

**Published:** 2021-10-30

**Authors:** Małgorzata Zimny, Małgorzata Starczewska, Małgorzata Szkup, Anna Cybulska, Elżbieta Grochans

**Affiliations:** 1Department of Obstetrics and Pathology of Pregnancy, Pomeranian Medical University, 71-210 Szczecin, Poland; mmzimny@wp.pl; 2Department of Nursing, Pomeranian Medical University, 71-210 Szczecin, Poland; m-lary@wp.pl (M.S.); anna.cybulska@pum.edu.pl (A.C.); grochans@pum.edu.pl (E.G.)

**Keywords:** body composition, perimenopausal period, lipid profile, glucose level, insulin, CRP, glycated haemoglobin, type 2 diabetes

## Abstract

Background and Objectives: The aim was to compare body composition and levels of biochemical blood parameters and identify relationships between biochemical parameters and body composition of women with type 2 diabetes and healthy ones, both in perimenopausal period (172 women aged between 45 and 65 come from the West Pomeranian Voivodeship, Poland). Materials and Methods: The study consisted of an interview, body composition analysis with Jawon Medical IOI-353 (Yuseong, South Korea) analyser and venous blood biochemical analysis (lipid profile, levels of glucose, insulin, CRP, glycated haemoglobin). Results: The vast majority of body composition measurements varied between study and control groups in a statistically significant way (*p* < 0.05) except protein and soft lean mass of the torso. Statistically significant differences between the two groups have been observed in case of all biochemical parameters (*p* < 0.001). Conclusions: Body composition of women suffering from type 2 diabetes significantly varied from body composition of healthy women. Results of the first group were characterised by higher values, especially in case of general parameters, abdominal area, content of adipose tissue and soft tissues. Relationship between body composition and biochemical results may be observed, especially in level of triglycerides, CRP and insulin. Higher concentrations of these parameters were associated with increased values of majority of body composition measurements regardless of type 2 diabetes incidence.

## 1. Introduction

The perimenopausal period is the time of numerous changes in a woman’s life—changes in biological, mental, social, and cultural spheres. Symptoms resulting from hormonal changes, caused by decreasing hormonal function of ovaries and a plummeting level of endogenous oestrogen, result in adverse changes in glucose and insulin metabolism, adipose tissue distribution, coagulation cascade disorders, and endothelium dysfunction. The severity and type of symptoms vary from individual to individual due to genetic predispositions and environmental factors. Vasomotor and psychosomatic symptoms appear. Hormonal disorders cause an increase in body mass, influence distribution of adipose tissue, reduce energy expenditure and decrease insulin secretion and sensitivity of cells to this hormone, all of which predispose to the development of type 2 diabetes [1,2,3,4]. Decreased sensitivity of cells to insulin is rather associated with adipose tissue distribution than its total mass. Fat in some people, not only obese ones, is deposited incorrectly and accumulates in the heart, skeletal muscles, blood vessels, and abdomen. Obesity is considered a main factor in the pathogenesis of cardiovascular, metabolic and neoplastic diseases [5,6,7]. Perimenopausal women were characterized not only by hormonal changes but also by various phenotypic and biochemical changes that could predispose them to the development of type 2 diabetes [8,9]. Premenopausal women had an increase in visceral fat storage from 5–8% of total body fat to 15–20% in the postmenopausal period and a decrease in muscle mass [2]. Lapidus et al. in their study showed a significant association between WHR and incidence of cardiovascular disease indicating fat distribution [10]. A study of Polish menopausal women, conducted by Weber-Rajek et al., showed low levels of maintenance of proper dietary habits and daily health practices related to sleep, rest and physical activity [11]. In Poland, in comparison to other European Union countries, we also observe a lower level of self-assessed health status among perimenopausal women, which, despite its subjective character, is considered an accurate predictor of mortality. Assessment of physical condition is also unsatisfactory compared to other countries [12].

In menopausal women lean body mass decreases with age and adipose tissue is re-distributed which results in an increase of visceral fat and waist circumference (WC) [13]. Visceral adipose tissue intensifies the production of proinflammatory cytokines, increases circulating plasma free fatty acids, and promotes synthesis of reactive oxygen species contributing to insulin resistance―the pathogenic cause for metabolic syndrome [14,15]. Even though the relationship between the menopausal state and decreased sensitivity to insulin has not been fully explained, it is believed that abdominal obesity might be a potential link between the menopausal period and the level of insulin resistance [16,17]. When metabolic disorders occur, the levels of postmeal glucose, C-reactive protein (CRP), and inter-leukin-6 (IL-6) increase; hypertriglyceridemia ensues and the levels of apolipoprotein A1 and HDL cholesterol decrease [18]. The generation and secretion of biologically active compounds by adipose tissue significantly influence cells’ lipid metabolism [19].

Oestrogen deficiency and increased levels of androgens may be associated with in-creased risk of cardiovascular diseases. After menopause ovaries still secrete androgens, bioavailability of which is greater due to decreased sex hormone binding protein (SHBG) concentration, which additionally increases insulin resistance [20,21]. Increased androgen levels have a highly aterogenous effect [22]. Oestrogen deficiency has a negative effect on lipid panel causing an increase in total cholesterol (TC), LDL cholesterol (especially low-density particles) and triglycerides (TG), and a drop in HDL cholesterol concentration [23,24,25].

The aim of this study was to compare body composition and the levels of biochemical blood parameters (lipid panel, glucose, insulin, CRP, glycated haemoglobin) as well as to identify relationships between chosen biochemical parameters and body composition of perimenopausal women with type 2 diabetes and healthy ones.

## 2. Materials and Methods

The study involved 172 perimenopausal women, aged between 45 and 65 years in 2018–2019. Women were recruited who represented the general population of the West Pomeranian Voivodeship in north-west Poland. The size of the study sample was established on the basis of statistical data concerning the size of the 45–64-year-old female population in the West Pomeranian Voivodeship in 2017 [26]. The confidence level was set at 95%, the maximum error at 7%, and the estimated fraction size at 0.5. The total number of women qualified for the study was 172. The respondents were divided into two groups: a study group consisting of 68 women with clinically diagnosed type 2 diabetes and a control group including 104 healthy women. The average age of the respondents was 54.9 years (study group—58.7, control group—52.3). 56.4% lived in a city with a population of over 100,000 people. The majority of the women had secondary (45.3%) or higher education (33.7%) (study group—mostly secondary education: 53.9%; control group—mostly higher education: 45.2%); 76.2% were married (study group—76.5%; control group—76%), and 75.0% were professionally active (study group—50%; control group—91.4%).

The study was survey-based and performed with the questionnaire technique. After that we analysed body composition using a body composition analyser Jawon Medical IOI-353 (Yuseong, South Korea). The research described in this manuscript is a part of a larger research project that aimed to determine the relationship between body composition and the functioning of women with type 2 diabetes [27]. In the next stage, blood was drawn in order to carry out laboratory analysis and mark biochemical parameters (lipid panel, levels of glucose, insulin, CRP, glycated haemoglobin).

### 2.1. Procedure of Body Composition Assessment

The measurements were carried out using body composition analyser Jawon Medical IOI-353 (Yuseong, South Korea) which holds EC0197 certificate and complies with the MDD 93/42EEC Directive concerning medical devices. The device uses bioelectric imped-ance, which is a reliable, safe, non-invasive, and above all, an effective way of examining individual components in human body. The anthropometric data was collected, visceral adipose tissue was assessed, and individual compartments, basal metabolic rate and metabolic age of examined women were determined with the analyser. The producer recommends that Jawon Medical IOI-353 is not used in pregnant patients and those with a pacemaker. All participants of the study were informed about contraindications for per-forming body analysis and confirmed their absence. The study was conducted in the morning; all patients had a fasting period of at least eight hours.

### 2.2. Procedure of Laboratory Tests Marking

10 mL of blood was drawn from a peripheral venous catheter using serum clot activator Vacutainer closed system in order to conduct biochemical analysis. Blood samples were collected in a treatment room in accordance with the rules and procedures regarding blood collection by qualified and competent staff. Biological material was stored and transported in compliance with current procedures. The examination was carried out in the certified laboratory of Professor Tadeusz Sokołowski Independent Public Clinical Hospital No 1 of the Pomeranian Medical University in Szczecin. Lipid profile, glucose, insulin, CRP, and glycated haemoglobin were marked in collected blood samples.

### 2.3. Organization and Course of Study

The study was carried out in Poland among women from the West Pomeranian Voivodeship. The study group included females aged between 45 and 65 years with diagnosed type 2 diabetes. The control group were healthy women aged between 45 and 65 years. The patients were informed about the study via leaflets and posters placed in public spaces such as town halls, schools, primary care centres, diabetes clinics and hospital wards. The study is a part of a greater project implemented by the study team.

The inclusion criteria for the study group were: diagnosed type 2 diabetes, and the lack of thyroid disorders, neoplastic and mental diseases, both currently and in the patient’s history. The exclusion criteria were using hormone replacement therapy (HRT) and hypolipidemic agents.

The inclusion criteria for the control group were the lack of diabetes, thyroid disorders, neoplastic and mental diseases, both currently and in the patient’s history. The exclusion criteria were diabetes, using HRT and hypolipidemic agents.

Each person, both from the study and the control group, who met the inclusion criteria received general information regarding the course and purpose of the study, as well as instructions on completing the questionnaires. After giving written consent to participate in the study, each patient received a questionnaire form. Afterwards, the patients underwent body composition analysis with Jawon Medical IOI-353 analyser (Yuseong, South Korea). The whole procedure was handled by trained and qualified staff. Each patient received a printout from the analyser with interpretation. The respondents were informed that the study is entirely voluntary and anonymous as well as that the results obtained will be used for scientific purposes and will not influence their therapeutic process.

The study was approved by the Bioethical Commission of the Pomeranian Medical University in Szczecin (no. KB-0012/90/15). It was conducted in accordance with the Declaration of Helsinki. Informed consent was obtained from all subjects involved in the study. Additionally, the participants were informed about the possibility of resignation at any stage without the need to provide a justification.

### 2.4. Statistical Analysis

The analysis of quantitative variables was conducted by calculating basic descriptive statistics: mean, standard deviation, median, quartiles, as well as minimum and maximum values. The analysis of qualitative variables was performed by calculating the number and percentage of occurrences of each value. Statistical inference was performed based on the assumption of normal distribution and the following statistical tests. The comparison of quantitative variables in two groups was performed using the Mann-Whitney U test. Normality of the variable distribution was assessed using the Shapiro-Wilk test. A significance level of 0.05 was adopted in the analysis. Thus, all *p* values below 0.05 were interpreted as indicating significant relationships.

In addition, measures of correlation were also used to show the degree of dependence between the variables analysed. The correlation between two quantitative variables was analysed using Spearman’s coefficient (when at least one of them did not have normal distribution).

The strength of the relationship was interpreted according to the following scheme:|r| ≥ 0.9—very strong relationship,0.7 ≤ |r| < 0.9—strong relationship,0.5 ≤ |r| < 0.7—moderately strong relationship,0.3 ≤ |r| < 0.5—weak relationship,|r| < 0.3—very weak (negligible) relationship.

The statistical analysis was performed in the R program, version 3.4.0.

## 3. Results

The average age of the surveyed was 54.9 years: in the group of women with type 2 diabetes 58.7 years and in the control group 52.3 years. Both groups were very similar in terms of the proportion of postmenopausal to premenopausal women. There were no statistically significant differences in this respect.

The women from both groups, i.e., the study and control groups, did not differ in terms of sociodemographic data. More than half of both groups (56.4%) lived in cities with over 100,000 residents, the majority of them had secondary (45.3%) and higher education (33.7%). 76.2% of the surveyed women were in a formal relationship and were employed (75%). The vast majority of the women (74.4%) were not menstruating. Among the women who were not menstruating an average time from the last menstruation was 9.79 years. In the group with type 2 diabetes, average duration of illness was 8.4 years. 89.7% of the women treated diabetes with oral agents, 20.6% used insulin, 8.8% used diet and physical activity, and 1.5% had an insulin pump. The study was conducted in two stages. 

There were statistically significant differences in body composition parameters between the study and the control group (*p* < 0.05), except for protein and the content of abdominal soft tissue. In the study group, the parameters were higher, especially body mass (M ± SD: 82.4 ± 14.9 vs. 72.1 ± 14.5) and adipose tissue mass (M ± SD: 32.8 ± 9 vs. 25.9 ± 8.6). Analysis of abdominal circumference showed that standard body mass was lower, and all the other parameters were higher than in the control group (M ± SD: 55 ± 7.7 vs. 58.1 ± 4.7). The distribution of fat tissue in torso (M ± SD: 16.8 ± 4.7 vs. 13.3 ± 4.4) and in both lower extremities (left M ± SD: 5.8 ± 1.7 vs. 4.7 ± 1.6; right M ± SD: 5.9 ± 1.6 vs. 4.7 ± 1.5) in both groups were the most diverse. Statistically significant differences were also found between both groups in terms of all parameters included in the guidebook of control and recommendations (*p* < 0.05). In the study group, impedance (electrical resistance of organ-ism’s tissues) was lower (M ± SD: 427.2 ± 60.4 vs. 475.4 ± 60.8) while metabolic age (M ± SD: 62.4 ± 6.3 vs. 53.6 ± 5.4), aim to control (quantity of kilograms to reduce) (M ± SD: 18.9 ± 9.1 vs. 12 ± 9.8) and duration of treatment (period during which patient was supposed to meet her target weight) (M ± SD: 37.7 ± 18.2 vs. 23.8 ± 16.4) were higher than in the control group. This manuscript is part of a larger study. Please, see the publication [27] for characteristics of the study and the control groups considering body composition analysis, control guides and recommendations.

Statistically significant differences between the study and the control group were established in all tested biochemical parameters (*p* < 0.001). In the study group, average levels of total cholesterol, HDL and LDL were lower and all other parameters were higher than in the control group. The levels of glucose and insulin were significantly different which indicates type 2 diabetes in the study group. Total cholesterol, triglyceride, CRP, and insulin levels were characterized by greatest diversity in both groups (Table 1).

In the study group, the level of HDL cholesterol significantly correlated with 21 parameters. Similarly, in the control group it corresponded with 25 parameters. Those correlations were negative–the higher the level of HDL cholesterol, the lower the values of analysed parameters, and the other way round–the higher the values of those parameters, the lower the level of HDL cholesterol. In the study group, the level of triglycerides significantly correlated with 25 parameters and in the control group with 27; it was a positive correlation. The only exception was a negative relationship with impedance in the control group (Table 2).

In the study group, the level of glucose significantly correlated with two parameters, namely standard body weight and metabolic age. The link between standard body weight was negative and positive with metabolic age. In the control group, the level of glucose significantly correlated with 18 parameters and those links were positive. The only exception was a negative correlation with impedance. In the study group, the level of glycated haemoglobin does not create a significant relationship with any of the parameters. However, in the control group it correlated with 25 parameters apart from protein, standard body weight, adipose tissue of right arm, basal metabolic rate, and estimated total energy expenditure, all of which were positive. The only exception was a negative correlation with impedance (Table 3).

In the study group, the level of insulin correlated with 23 parameters and with 28 parameters in control group except for standard body mass and estimated total energy expenditure. All those correlations were positive. The only exception was a negative correlation with impedance in the control group. In the study group, CRP level significantly correlated with 17 parameters and with 28 in the control group with the exception of standard body weight and estimated total energy expenditure. Those correlations were positive. The only exception was a negative correlation with impedance in the control group (Table 4).

## 4. Discussion

The prevalence rate of diabetes is highest in Western countries, and WHO forecasts predict a further increase in its incidence worldwide—by 2025, the number of people with diabetes in economically and socially developed countries will have increased by 43% [28]. A significant increase in the incidence of type 2 diabetes is fostered by the coexistence of overweight or obesity, poor diet and low physical activity. 

Our research indicated that 74.4% of the surveyed women in both groups were overweight or obese, but more obese women belonged to the group with diagnosed type 2 diabetes. In Poland, about 60% of women after menopause suffer from excessive weight and obesity [29]. Moreover, the percentage of obese women double after menopause and visceral adipose tissue mass increases over 50% [30]. Also in other countries of the world, women over 40 years of age are overweight and obese–61% of Spanish women [31] and 66% of Brazilian ones [32]. Obesity and excessive body weight are also a problem of diabetic patients, which was confirmed in our study. In Kudaj-Kurowska et al.’s study, 93% of the surveyed women had improper body weight, 68% of whom were obese [33]. This phenomenon is also observed in patients with newly diagnosed diabetes. Nationwide study ARETAEUS1 indicated that 37.4% of newly diagnosed diabetics were overweight and 51.9% were obese [34]. Diet is considered an integral part of diabetes treatment. Proper nutrition causes blood glucose levels to be as close to normal as possible, achieving optimal serum lipids, reduces and then maintains optimal body weight in overweight and obese people, thus improving overall health. According to the guidelines of the Polish Diabetes Association, the diet of a diabetic patient should not deviate from the basic recommendations for healthy eating, however, individual energy and nutritional indicators of the diet should be tailored to the individual needs of each patient, taking into account sex, age, health status, physical effort performed and metabolic status of the patient (glycemia, HbA concentration, lipid profile, kidney status, blood pressure values and other parameters) [35,36]. For obese diabetics, reduced energy intake and weight loss (0.5 kg per week) are recommended. A convenient way to plan a diabetic’s diet is to know and use the carbohydrate exchangers to calculate the carbohydrate content of different foods. One carbohydrate exchanger is the amount of a given food product in which 10 g of carbohydrates are contained. The system of exchangers enables the introduction of various products into the diet while maintaining the desired constant level of carbohydrate content in individual meals, allowing it to be diversified. A properly composed diet, in addition to the appropriate amount of carbohydrates, must provide the full demand for energy and all the ingredients necessary for the proper functioning of the body [28,37]. Also physical activity is an important part of the therapeutic management of diabetes as it, along with diet therapy, promotes normalization of body weight [38]. Systematic exercise is a very important way to improve your health. Prevention and treatment of chronic non-communicable diseases resulting from malnutrition and low physical activity is currently one of the main priority actions taken by the European Union. 

Reports on the Polish population indicate that the majority of women and men declare a passive way of spending their free time [38]. Therefore, programs for various forms of physical activity in patients should be constantly implemented, because most of the respondents show a rather passive way of rest, and regular physical activity is a prophylaxis not only of diabetes, but as many as 35 various chronic diseases [39].

Our study showed statistically significant differences within all evaluated parameters. In women with type 2 diabetes, the presence of visceral adipose tissue and increased abdominal circumference were observed. Metanalysis conducted by Ambikairajah et al. demonstrated an increase in central adipose tissue and a decrease in the percentage of adipose tissue in women’s legs in the postmenopausal period [13]. In Toth et al.’s study, women after menopause had 36% of visceral adipose tissue and 49% of visceral fat surface more than their premenopausal counterparts [40]. Also, in Pachocka’s studies central obesity prevailed, especially in the perimenopausal women. However, no statistically significant differences in body composition were established [41]. It is worthwhile to make a comparison between the group of perimenopausal women included in our study and the group of premenopausal women. A study conducted by Yeung et al. among premenopausal African-American and European-American women indicated that the ratio of trunk fat to limb fat is significantly inversely correlated with sex-hormone binding globulin, which is associated with increased risk of type 2 diabetes after controlling for homeostasis model assessment–insulin resistance, estradiol, physical activity, and caloric intake [42]. Intra-abdominal adipose tissue is an important fat depot that is associated with an abnormal metabolic profile [43] and significantly associated with sex-hormone binding globulin independent of insulin in a female population [44]. Despite these reports, the associations among intra-abdominal adipose tissue, insulin, and sex-hormone binding globulin in premenopausal women remain unclear.

It is emphasized that although the relationship between body fat mass and the risk of type 2 diabetes, dyslipidemia, hypertension, and cancer is crucial, it is visceral fat that appears to have a greater negative impact on this risk than the amount of total fat [45]. On the other hand, it cannot be ruled out that in addition to central obesity, adipose tissue in other areas of a woman’s body may play an important role in the development of diabetes. It has been observed that an increased risk of type 2 diabetes positively correlates with a patient’s bust size [46]. According to Groop and Orho-Melander, both the amount of visceral adipose tissue and central obesity significantly correlate with insulin resistance [47]. 

In addition to the tendency to gain weight, perimenopausal women are at risk of changes in lipid metabolism due to oestrogen deficiency, which causes an increase in the levels of total cholesterol and lipoproteins, thus contributing to the development of atherosclerosis [48]. A study by Przysławski et al. showed that total cholesterol and LDL cholesterol significantly increase during menopause, and the changes progress with age [49]. Besides the increase in serum cholesterol and low density lipoprotein (LDL) levels, triglyceride (TG) levels increase and HDL levels significantly decrease [50]. Matthews et al.’s study showed an average 10–20% increase in LDL compared to the pre-menopausal period [51]. Central obesity in the menopausal period significantly contributes to lipid disorders. The study by Franiak-Pietryga et al. showed that serum lipid levels (LDL cholesterol fraction, TG) were significantly higher. However, the concentration of LDL cholesterol fraction was lower than in the control group [52].

In the study by Piskorz et al., postmenopausal women had significantly higher mean levels of total cholesterol, LDL cholesterol, and triglycerides compared with menstruating women [53]. Similar results were obtained by Shende et al. [54] and Gower et al. [55], who analysed pre- and postmenopausal women. Total cholesterol and LDL cholesterol levels were substantially higher in postmenopausal women. Additionally, Shende et al. observed that HDL cholesterol levels were significantly lower in postmenopausal women. On the other hand, in the study by Milewicz et al., healthy postmenopausal women showed a decrease in total cholesterol and LDL cholesterol with a simultaneous increase in HDL cholesterol after continuous transdermal use of 5 mg of dydrogesterone per 24 h [56].

Type 2 diabetes is regarded as one of the most common diseases of affluence. Any chronic disease increases stress, which is associated with increased blood glucose levels [57]. In the course of diabetes, one of the greatest difficulties is achieving the goal of lipid control. In our own study, in the group of women with type 2 diabetes, mean levels of total cholesterol, HDL and LDL cholesterol were lower than in the control group. Due to the health condition, the concentration of glucose and insulin in the studied women differed significantly. Kudaj-Kurowska’s team found in their study that an unsatisfactory number of patients with type 2 diabetes met the criteria for compensated diabetes recommended by the Polish Diabetes Association. Only 38% of the subjects achieved the recommended LDL-c level, and 41% reached the target for total cholesterol. The desired HDL-c concentration was achieved by 63% of women and 57% of men [33]. However, in the study by Dudzińska et al., 40.4% of patients after insulin treatment achieved the target levels of LDL-c and total cholesterol [58]. 

The analysis by Arsenault et al. showed a relationship between the phenotype of hypertriglyceridemia (increased waist circumference and TG concentration) and the risk of cardiovascular events. People with increased waist circumference and raised triglycerides had a higher average body mass index than those with shorter waist circumference and lower TG levels. There were also higher mean blood pressure values, increased levels of CRP, total and LDL cholesterol, and lower levels of HDL, which worsened the cardiometabolic profile [59]. The study by Tao et al. revealed a positive correlation between TG and BMI [60], which was also confirmed in our analysis. Other researchers have argued that the combination of an increased waist circumference and an elevated level of TG is associated with aortic calcification in postmenopausal women [61]. Close observation of patients with high TG levels and BMI is particularly important, especially in the case of women, as both of these factors are closely related to the development of hypertension, diabetes, cardiovascular diseases and other metabolic diseases [60].

Decreased sex hormone levels and abdominal obesity may exacerbate inflammatory processes. One of the inflammatory process indicators analysed in this study is CRP. Our study showed that higher CRP levels are associated with increased values of most body composition parameters both in women with type 2 diabetes and in healthy controls. Similar results were obtained by Stefańska et al.―their analysis of peri- and postmenopausal women demonstrated a significant correlation between body mass indices (BMI, WHR, WC) and CRP concentration only in postmenopausal women [62]. Karolkiewicz’s team studied elderly women and found that higher lipid values and CRP levels in healthy overweight and obese women supported the researchers’ hypothesis that weight gain is associated with risk of atherosclerosis regardless of age [63]. Piskorz et al. observed that CRP levels were significantly higher in postmenopausal women than in menstruating ones [53]. A study of healthy postmenopausal women in the Women’s Health Study showed that healthy women with high hs-CRP protein levels (assessed by high-sensitivity methods) were 4.4 times more likely to experience cardiovascular events compared with patients with lower hs-CRP protein levels [64]. In their analysis of the Chinese population, Hong et al. found a positive relationship between hs-CRP and risk of developing metabolic syndrome, but only in women [65]. They also found that high CRP levels multiply the risk of ovarian cancer, especially in overweight and obese women [66].

Elevated CRP, symptomatic of obesity, may signify an overproduction of proinflammatory cytokines that directly impinge on the insulin signal transduction pathway. Amullah et al., who studied subjects of different ages, observed significantly increased glucose and glycated hemoglobin (HbA1c) levels, and significantly higher insulin resistance-HOMA and BMI values in both diabetic patients and subjects with elevated hs-CRP levels [67]. However, our own study showed that significantly higher concentrations of all studied parameters were observed only in women with type 2 diabetes.

The main strength of our study is its originality and objectivity, obtained by using a standardized research tool and a professional body composition analyser. It is difficult to find in the literature a similar analysis in relation to perimenopausal women, especially considering the presence of type 2 diabetes.

### Limitations

Our study has some limitations. There was a limited number of studies concerning the relationship between the characteristics of type 2 diabetes (disease duration, treatment methods), body composition, and biological function. Next, a larger number of participants, especially those diagnosed with type 2 diabetes, would likely increase the reliability of the study conducted. Finally, the difference in mean age between the two groups (58.7 vs. 52.3) may be a potentially confounding factor. The results obtained, while promising, require replication to be applied to a wider population, 

A practical dimension of the presented research may be the proposal to introduce additional examinations for perimenopausal women, based on the analysis of body composition, which will undoubtedly contribute to better planning of preventive measures in this age group.

## 5. Conclusions

The functioning of perimenopausal women with and without type 2 diabetes differed in terms of biological and biochemical parameters.

The body composition of women with type 2 diabetes differed significantly from that of healthy women. The results of the first group were characterized by higher values, especially for general parameters, abdominal area, body fat and soft tissue content.

A relationship can be observed between body composition and biochemical test results, which is especially true for the levels of triglycerides, CRP, and insulin. Higher concentrations of these parameters were associated with elevated values of the majority of body composition measurements, irrespective of type 2 diabetes. There was no link between body composition and the levels of glucose and glycated haemoglobin in women with type 2 diabetes, whereas such an association was observed in healthy women.

## Figures and Tables

**Table 1 ijerph-18-11422-t001:** Characteristics of the study and control groups in terms of biochemical results.

Parameters	Group	*n*	M ± SD	Me	Min-Max	Q_1_–Q_3_	*p*
Total cholesterol [mg/dL]	Study	68	183.3 ± 42.5	182	94–289	152.75–221.5	<0.001
	Control	104	207.8 ± 37.9	207.8	119–311	183–233	
HDL [mg/dL]	Study	68	55.4 ± 16.2	53	5–97	45.75–64.3	<0.001
	Control	104	67.0 ± 18.4	64	33–143	54.53–78	
LDL [mg/dL]	Study	67	98.3 ± 36.5	90.2	30.2–190.6	66.4–127.6	<0.001
	Control	103	121.6 ± 35.5	119.8	38.2–222.8	98.1–140.3	
Triglycerides [mg/dL]	Study	68	149.1 ± 76.5	127	48–445	98–171.6	<0.001
	Control	104	98.2 ± 57.8	84	28–417	64.75–120	
Fasting Blood Glucose [mg/dL]	Study	68	138.3 ± 53.4	127	80–348	102.75–150.6	<0.001
	Control	104	84.7 ± 19.7	82	63–254	74–92	
CRP [mg/L]	Study	68	5.6 ± 10.3	2.6	1–76.5	1.75–4.9	<0.001
	Control	104	2.4 ± 2.9	1.3	1–20	1–2.5	
HbA1c [%]	Study	67	6.8 ± 1.5	6.56	5–13.15	5.92–7.4	<0.001
	Control	104	5.3 ± 0.6	5.29	2.9–10.2	5.07–5.5	
Insulin [µIU/mL]	Study	67	20.7 ± 18.6	17.3	3.5–148.2	10.55–26.1	<0.001
	Control	104	9.9 ± 5.6	8.7	2.3–32.2	6.38–11.4	

*n*-number. M-mean. SD-standard deviation. Me-median. Q1-the first quartile. Q3-the third quartile. Mann-Whitney U test. *p*-level of statistical significance.

**Table 2 ijerph-18-11422-t002:** Correlation of body composition with levels of HDL cholesterol and in study and control groups.

Parameters	Level of HDL Cholesterol				Level of Triglycerides			
	Study Group		Control Group		Study Group		Control Group	
	Correlation Coefficient	*p*	Correlation Coefficient	*p*	Correlation Coefficient	*p*	Correlation Coefficient	*p*
Weight [kg]	−0.284	0.019 *	−0.271	0.005 **	0.34	0.005 **	0.408	<0.001 ***
L.B.M. [kg]	−0.267	0.028 *	−0.312	0.001 **	0.321	0.008 **	0.362	<0.001 ***
M. B. F. [kg]	−0.275	0.023 *	−0.282	0.004 **	0.357	0.003 **	0.403	<0.001 ***
S.L.M. [kg]	−0.275	0.023 *	−0.309	0.001 **	0.332	0.006 **	0.35	<0.001 ***
Mineral [kg]	−0.287	0.018 *	−0.302	0.002 **	0.351	0.003 **	0.393	<0.001 ***
Protein [kg]	−0.243	0.046 *	−0.308	0.001 **	0.293	0.015 *	0.308	0.001 **
T.B.W. [kg]	−0.254	0.036 *	−0.31	0.001 **	0.302	0.012 *	0.361	<0.001 ***
P.B.F. [%]	−0.17	0.166	−0.195	0.047 *	0.297	0.014 *	0.35	<0.001 ***
B.M.I. [kg/m^2^]	−0.239	0.05	−0.229	0.019 *	0.299	0.013 *	0.375	<0.001 ***
Fatness [%]	−0.308	0.104	---	---	0.289	0.128	---	---
Level [steps]	−0.172	0.162	−0.187	0.057	0.275	0.023 *	0.352	<0.001 ***
V.F.A. [cm^2^]	−0.164	0.183	−0.196	0.046 *	0.289	0.017 *	0.36	<0.001 ***
A.C. [cm]	−0.276	0.023 *	−0.274	0.005 **	0.358	0.003 **	0.405	<0.001 ***
W.H.R.	−0.159	0.195	−0.193	0.049 *	0.287	0.018 *	0.349	<0.001 ***
Std. Mc. [kg]	−0.02	0.87	−0.183	0.065	0.111	0.369	0.083	0.409
L. arm–M.B.F. [kg]	−0.295	0.015 *	−0.277	0.004 **	0.39	0.001 **	0.417	<0.001 ***
L. arm-S.L.M. [kg]	−0.282	0.02 *	−0.283	0.004 **	0.212	0.082	0.282	0.004 **
R. arm-M.B.F. [kg]	−0.299	0.013 *	−0.28	0.004 **	0.406	0.001 ***	0.426	<0.001 ***
R. arm-S.L.M. [kg]	−0.207	0.09	−0.274	0.005 **	0.223	0.067	0.266	0.006 **
Trunk-M.B.F. [kg]	−0.262	0.031 *	−0.27	0.006 **	0.345	0.004 **	0.399	<0.001 ***
Trunk-S.L.M. [kg]	−0.243	0.046 *	−0.309	0.001 **	0.341	0.004 **	0.366	<0.001 ***
L. leg-M.B.F. [kg]	−0.257	0.034 *	−0.274	0.005 **	0.341	0.004 **	0.405	<0.001 ***
L. leg-S.L.M. [kg]	−0.225	0.066	−0.293	0.003 **	0.317	0.009 **	0.325	0.001 ***
R. leg-M.B.F. [kg]	−0.283	0.019 *	−0.26	0.008 **	0.361	0.002 **	0.375	<0.001 ***
R. leg-S.L.M. [kg]	−0.256	0.035 *	−0.31	0.001 **	0.298	0.014 *	0.335	0.001 ***
B.M.R. [kcal]	−0.241	0.048 *	−0.304	0.002 **	0.368	0.002 **	0.325	0.001 ***
T.E.E. [kcal]	−0.24	0.048 *	−0.16	0.104	0.334	0.005 **	0.174	0.077
A.M.B. [years]	−0.08	0.517	0.008	0.939	−0.047	0.703	0.111	0.26
Impedance [Ω]	0.239	0.049 *	0.146	0.139	−0.148	0.229	−0.213	0.03 *
Target to control [kg]	−0.261	0.032 *	−0.209	0.034 *	0.348	0.004 **	0.375	<0.001 ***
Therapy duration [weeks]	−0.261	0.032 *	−0.235	0.016 *	0.35	0.003 **	0.394	<0.001 ***

Spearman’s rank correlation coefficient. L.B.M.-Lean Body Mass [kg]; S.L.M.-Soft Lean Mass [kg]; M.B.F.–Mass of Body Fat [kg]; T.B.W.-Total Body Water [%]; P.B.F.-Per cent of Body Fat [%]; B.M.I.-Body Mass Index [kg/m^2^]; V.F.A.-Visceral Fat Area [cm^2^]; A.C.-Abdominal Circumference [cm]; W.H.R.-Waist Hip Ratio; Std. Mc.–standard body mass [kg]; B.M.R.-Basal Metabolic Rate [kcal]; T.E.E.-Total Energy Expenditure [kcal]; A.M.B.-Age Matched of Body [years]; *** *p* < 0.001; ** *p* < 0.01; * *p* < 0.05.

**Table 3 ijerph-18-11422-t003:** Correlation of body composition with levels of fasting blood glucose and HbA1c in study and control groups.

	Level of Fasting Blood Glucose				Level of HbA1c			
	Study Group		Control Group		Study Group		Control Group	
	Correlation Coefficient	*p*	Correlation Coefficient	*p*	Correlation Coefficient	*p*	Correlation Coefficient	*p*
Weight [kg]	0.087	0.481	0.219	0.026 *	0.029	0.814	0.415	<0.001 ***
L.B.M. [kg]	−0.023	0.85	0.167	0.091	−0.054	0.666	0.25	0.01 *
M.B.F. [kg]	0.146	0.236	0.244	0.013 *	0.097	0.436	0.451	<0.001 ***
S.L.M. [kg]	−0.02	0.87	0.149	0.132	−0.044	0.723	0.229	0.019 *
Mineral [kg]	0.078	0.528	0.232	0.018 *	0.022	0.859	0.379	<0.001 ***
Protein [kg]	−0.084	0.498	0.11	0.265	−0.084	0.5	0.144	0.144
T.B.W. [kg]	0.04	0.743	0.164	0.096	−0.015	0.901	0.25	0.011 *
P.B.F. [%]	0.159	0.196	0.218	0.026 *	0.114	0.357	0.452	<0.001 ***
B.M.I. [kg/m^2^]	0.181	0.14	0.255	0.009 **	0.125	0.314	0.437	<0.001 ***
Fatness [%]	0.081	0.678	---	---	0.123	0.525	---	---
Level [steps]	0.204	0.096	0.229	0.02 *	0.156	0.209	0.461	<0.001 ***
V.F.A. [steps]	0.167	0.174	0.228	0.02 *	0.122	0.325	0.467	<0.001 ***
A.C. [cm]	0.145	0.238	0.244	0.012 *	0.096	0.439	0.45	<0.001 ***
W.H.R.	0.169	0.168	0.226	0.021 *	0.119	0.338	0.467	<0.001 ***
Std. Mc. [kg]	−0.244	0.045 *	−0.078	0.437	−0.205	0.096	−0.092	0.358
L. arm-M.B.F. [kg]	0.123	0.318	0.244	0.013 *	0.034	0.784	0.454	<0.001 ***
L. arm-S.L.M. [kg]	0.036	0.773	0.142	0.152	−0.006	0.959	0.207	0.035 *
R. arm-M.B.F. [kg]	0.086	0.486	0.256	0.009 **	0.001	0.992	0.454	<0.001 ***
R. arm-S.L.M. [kg]	0.064	0.605	0.124	0.21	0.071	0.567	0.19	0.053
Trunk-M.B.F. [kg]	0.147	0.231	0.238	0.015 *	0.101	0.414	0.439	<0.001 ***
Trunk-S.L.M. [kg]	−0.086	0.488	0.129	0.19	−0.088	0.477	0.213	0.03 *
L. leg-M.B.F. [kg]	0.14	0.254	0.244	0.012 *	0.126	0.31	0.444	<0.001 ***
L. leg-S.L.M. [kg]	0.027	0.829	0.142	0.15	−0.004	0.977	0.225	0.022 *
R. leg-M.B.F. [kg]	0.148	0.229	0.226	0.021 *	0.091	0.463	0.432	<0.001 ***
R. leg-S.L.M. [kg]	0.026	0.832	0.176	0.073	0.019	0.881	0.252	0.01 **
B.M.R. [kcal]	−0.099	0.42	0.063	0.528	−0.097	0.433	0.097	0.329
T.E.E. [kcal]	−0.068	0.584	0.057	0.566	−0.02	0.872	0.075	0.446
A.M.B. [years]	0.256	0.035 *	0.231	0.018 *	0.145	0.241	0.465	<0.001 ***
Impedance [Ω]	−0.177	0.148	−0.217	0.027 *	−0.157	0.205	−0.251	0.01 *
Target to control [kg]	0.179	0.143	0.247	0.011 *	0.125	0.314	0.474	<0.001 ***
Therapy duration [weeks]	0.178	0.145	0.272	0.005 **	0.126	0.308	0.454	<0.001 ***

Spearman’s rank correlation coefficient. L.B.M.-Lean Body Mass [kg]; S.L.M.-Soft Lean Mass [kg]; M.B.F.–Mass of Body Fat [kg]; T.B.W.-Total Body Water [%]; P.B.F.-Per cent of Body Fat [%]; B.M.I.-Body Mass Index [kg/m^2^]; V.F.A.-Visceral Fat Area [cm^2^]; A.C.-Abdominal Circumference [cm]; W.H.R.-Waist Hip Ratio; Std. Mc.–standard body mass [kg]; B.M.R.-Basal Metabolic Rate [kcal]; T.E.E.-Total Energy Expenditure [kcal]; A.M.B.-Age Matched of Body [years]; *** *p* < 0.001; ** *p* < 0.01; * *p* < 0.05.

**Table 4 ijerph-18-11422-t004:** Correlation of body composition with level of insulin and CRP in study and control groups.

Parameters	Level of Insulin				Level of CRP			
	Study Group		Control Group		Study Group		Control Group	
	Correlation Coefficient	*p*	Correlation Coefficient	*p*	Correlation Coefficient	*p*	Correlation Coefficient	*p*
Weight [kg]	0.393	0.001 ***	0.456	<0.001 ***	0.283	0.019 *	0.507	<0.001 ***
L.B.M. [kg]	0.267	0.029 *	0.369	<0.001 ***	0.205	0.094	0.377	<0.001 ***
M.B.F. [kg]	0.419	<0.001 ***	0.518	<0.001 ***	0.303	0.012 *	0.518	<0.001 ***
S.L.M. [kg]	0.26	0.034 *	0.349	<0.001 ***	0.209	0.088	0.357	<0.001 ***
Mineral [kg]	0.372	0.002 **	0.482	<0.001 ***	0.268	0.027 *	0.48	<0.001 ***
Protein [kg]	0.172	0.164	0.281	0.004 **	0.151	0.22	0.29	0.003 **
T.B.W. [kg]	0.297	0.015 *	0.369	<0.001 ***	0.227	0.063	0.377	<0.001 ***
P.B.F. [%]	0.472	<0.001 ***	0.51	<0.001 ***	0.274	0.024 *	0.498	<0.001 ***
B.M.I. [kg/m^2^]	0.413	0.001 ***	0.533	<0.001 ***	0.301	0.013 *	0.487	<0.001 ***
Fatness [%]	0.311	0.107	---	---	0.308	0.104	---	---
Level [steps]	0.45	<0.001 ***	0.526	<0.001 ***	0.291	0.016 *	0.505	<0.001 ***
V.F.A. [cm^2^]	0.469	<0.001 ***	0.519	<0.001 ***	0.271	0.025 *	0.501	<0.001 ***
A.C. [cm]	0.42	<0.001 ***	0.529	<0.001 ***	0.303	0.012 *	0.522	<0.001 ***
W.H.R.	0.459	<0.001 ***	0.522	<0.001 ***	0.258	0.034 *	0.489	<0.001 ***
Std. Mc. [kg]	−0.101	0.414	−0.046	0.644	−0.033	0.79	0.01	0.919
L. arm-M.B.F. [kg]	0.427	<0.001 ***	0.528	<0.001 ***	0.309	0.01 *	0.51	<0.001 ***
L. arm-S.L.M. [kg]	0.245	0.046 *	0.357	<0.001 ***	0.175	0.153	0.373	<0.001 ***
R. arm-M.B.F. [kg]	0.429	<0.001 ***	0.523	<0.001 ***	0.318	0.008 **	0.501	<0.001 ***
R. arm-S.L.M. [kg]	0.243	0.048 *	0.363	<0.001 ***	0.177	0.148	0.371	<0.001 ***
Trunk-M.B.F. [kg]	0.405	0.001 ***	0.527	<0.001 ***	0.309	0.01 *	0.524	<0.001 ***
Trunk-S.L.M. [kg]	0.233	0.058	0.353	<0.001 ***	0.188	0.125	0.355	<0.001 ***
L. leg-M.B.F. [kg]	0.409	0.001 ***	0.528	<0.001 ***	0.281	0.02 *	0.522	<0.001 ***
L. leg-S.L.M. [kg]	0.234	0.056	0.365	<0.001 ***	0.24	0.049 *	0.377	<0.001 ***
R. leg-M.B.F. [kg]	0.42	<0.001 ***	0.529	<0.001 ***	0.306	0.011 *	0.534	<0.001 ***
R. leg-S.L.M. [kg]	0.269	0.027 *	0.357	<0.001 ***	0.219	0.073	0.349	<0.001 ***
B.M.R. [kcal]	0.247	0.044 *	0.31	0.001 **	0.18	0.142	0.313	0.001 **
T.E.E. [kcal]	0.204	0.098	0.187	0.058	0.155	0.206	0.105	0.287
A.M.B. [lata]	0.111	0.372	0.263	0.007 **	0.126	0.306	0.232	0.018 *
Impedance [Ω]	−0.178	0.149	−0.318	0.001 ***	−0.186	0.129	−0.242	0.013 *
Target to control [kg]	0.418	<0.001 ***	0.552	<0.001 ***	0.296	0.014 *	0.485	<0.001 ***
Therapy duration [weeks]	0.421	<0.001 ***	0.542	<0.001 ***	0.295	0.014 *	0.533	<0.001 ***

Spearman’s rank correlation coefficient. L.B.M.-Lean Body Mass [kg]; S.L.M.-Soft Lean Mass [kg]; M.B.F.–Mass of Body Fat [kg]; T.B.W.-Total Body Water [%]; P.B.F.-Per cent of Body Fat [%]; B.M.I.-Body Mass Index [kg/m^2^]; V.F.A.-Visceral Fat Area [cm^2^]; A.C.-Abdominal Circumference [cm]; W.H.R.-Waist Hip Ratio; Std. Mc.–standard body mass [kg]; B.M.R.-Basal Metabolic Rate [kcal]; T.E.E.-Total Energy Expenditure [kcal]; A.M.B.-Age Matched of Body [years]; *** *p* < 0.001; ** *p* < 0.01; * *p* < 0.05.

## Data Availability

The data presented in this study are available on request from the corresponding author.

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
