# Peer review of "Body Composition and Biological Functioning in Polish Perimenopausal Women with Type 2 Diabetes"

_ijerph, 2021, doi:10.3390/ijerph182111422_

Round 1

Reviewer 1 Report

Thank you for your interesting manuscript! However, I have some suggestions still for it.

  1. The authors missed the method used for this study throughout the materials and methods.
  2. The results go to the biochemical markers without introducing the socio-demographic background of the participants. It should be known the background characteristic of the participants for checking up the bias, confounding and might have more discussion points for authors. This would be the limitation of the study if you do not have any data regarding with socio-demographic status of the participants.
  3. The first paragraph of the discussion is more relevant to the introduction. The introduction itself should be added more information like this actually.
  4. The discussion should be made more logical explanation based on your results rather than the current presentation. In addition, you should discuss the results on the background of Polish diet pattern, physical activity, etc, as far as you know type 2 diabetes is sophisticated concerning causal relationship.
  5. The conclusion should be brief on your findings and discussion, and give the recommendation for policy makers and further research needed. The second paragraph in the conclusion is not too much informative as a conclusion.
  6. The generalizability of the study must be reported in the manuscript.

Author Response

Dear Sir or Madam,

We are very grateful for the review of our article titled “Body composition and biological functioning in perimenopausal women with type 2 diabetes”. We would like to thank you for all your comments and suggestions, which helped us to improve our manuscript. We are grateful for good words about the general value of the manuscript, it is very motivating for us.

The following corrections have been introduced in order to address the suggestions of the Reviewer 1 (marked in the manuscript with blue):

  1. The authors missed the method used for this study throughout the materials and methods.

Thank you for this correction. We have corrected the description of Materials and Methods section: “The method of the diagnostic survey was used to conduct the study, with the use of the questionnaire technique. After that we made an analysis of body composition with body composition analyser Jawon Medical IOI-353 (Yuseong, South Korea). In the second stage, blood was drawn in order to carry out laboratory analysis and mark biochemical parameters (lipid panel, levels of glucose, insu-lin, CRP, glycated haemoglobin).”

  1. The results go to the biochemical markers without introducing the socio-demographic background of the participants. It should be known the background characteristic of the participants for checking up the bias, confounding and might have more discussion points for authors. This would be the limitation of the study if you do not have any data regarding with socio-demographic status of the participants.

We added an information: The average age of respondents was 54.9 years (study group 58.7, control group 52.3), 56.4% lived in a city with a population greater than 100 thousand. The majority had secondary (45.3%) or higher education (33.7%) (study group: the most popular was secondary education 53.9%; control group: higher education 45.2%); 76.2% were married (study group: 76,5%; control group 76%) and 75.0% professionally active (study group: 50%; study group 91.4%).

  1. The first paragraph of the discussion is more relevant to the introduction. The introduction itself should be added more information like this actually.

               We moved the content of the first paragraph of the discussion to the introduction and we have compiled a list of references: “Women in perimenopausal period not only presented with hormonal changes but also various phenotypic and biochemical alternations which might have had predisposed them to a development type 2 diabetes. Visceral fat storing increased from 5-8% of total adipose tissue in premenopausal period to 15-20% in postmenopausal stage and a decrease in muscle mass occurred.. Lapidus et al. in their studies showed a significant link between WHR and incidence of cardiovascular diseases pointing to the distribution of adipose tissue. The Weber-Rajek et al.’s study conducted among Polish menopausal women showed a low level of maintaining proper eating habits and daily health practices related to sleep, recreation, and physical activity. In Poland, compared to other Euro-pean Union countries, we also observe a lower self-reported health status among peri-menopausal women, which, despite the subjective character of such assessment, is con-sidered an accurate predictor of mortality. The assessment of physical condition is also unsatisfactory compared to other countries.”

  1. The discussion should be made more logical explanation based on your results rather than the current presentation. In addition, you should discuss the results on the background of Polish diet pattern, physical activity, etc, as far as you know type 2 diabetes is sophisticated concerning causal relationship.

            Thank you for this comment. We completed the discussion section with following fragments: ”Wskaźnik chorobowości z powodu cukrzycy jest najwyższy w krajach zachodnich, a prognozy WHO przewidują dalsze zwiększanie częstotliwości jej występowania na świecie – do 2025 roku liczba osób z cukrzycą w krajach gospodarczo i społecznie rozwiniętych wzrośnie o 43%. Znacznemu wzrostowi zachorowalności na cukrzycę typu 2, sprzyja współistnienie nadwagi lub otyłości, niewłaściwy sposób odżywiania i mała aktywność fizyczna.” and:

„Dieta jest uznawana za nieodłączny element leczenia cukrzycy. Prawidłowy sposób żywienia powoduje uzyskanie stężenia glukozy we krwi możliwie najbardziej zbliżonej do wartości prawidłowych, osiągnięcie optymalnych wartości lipidów w surowicy krwi, u osób z nadwagą i otyłością redukuje, następnie utrzymuje optymalną masę ciała tym samym poprawiając ogólny stan zdrowia. Zgodnie z wytycznymi Polskiego Towarzystwa Diabetologicznego dieta osoby chorej na cukrzycę nie powinna odbiegać od podstawowych zaleceń zdrowego żywienia, jednak poszczególne wskaźniki energetyczne i odżywcze diety powinny być dopasowane do indywidualnych potrzeb każdego pacjenta, uwzględniając płeć, wiek, stan zdrowia, podejmowany wysiłek fizyczny wykonywany, czy stan metaboliczny chorego (glikemia, stężenie HbA, lipidogram, stan nerek, wartości ciśnienia tętniczego i inne parametry). Chorym na cukrzycę z towarzyszącą otyłością zalecana jest zmniejszona podaż energii i utrata masy ciała (0,5 kg na tydzień). Wygodnym sposobem planowania diety chorego na cukrzycę jest poznanie i stosownie wymienników węglowodanowych, służących do obliczania zawartości węglowodanów w różnych produktach. Jeden wymiennik węglowodanowy, to taka ilość danego produktu żywnościowego, w jakiej zawarte jest 10 g węglowodanów. System wymienników umożliwia wprowadzenie do diety różnych produktów przy utrzymaniu na pożądanym stałym poziomie zawartości węglowodanów w poszczególnych posiłkach, umożliwiając jej urozmaicenie. Prawidłowo skomponowana dieta oprócz odpowiedniej ilości węglowodanów, musi zapewniać pełne zapotrzebowanie na energię i wszystkie składniki niezbędne dla prawidłowego funkcjonowania organizmu. Również aktywność fizyczna stanowi ważny element postępowania terapeutycznego w cukrzycy, gdyż wraz z dietoterapią sprzyja normalizacji masy ciała. Systematyczne podejmowanie wysiłku fizycznego jest bardzo ważnym sposobem na poprawę stanu zdrowia. Zapobieganie i leczenie przewlekłych chorób niezakaźnych wynikających z nieprawidłowego żywienia i małej aktywności fizycznej jest obecnie jednym z głównych działań priorytetowych podejmowanych przez Unię Europejską. Doniesienia z populacji polskiej wskazują, że większość kobiet i mężczyzn deklaruje bierny sposób spędzania wolnego czasu. Dlatego należy stale wdrażać programy różnych form aktywności fizycznej u pacjentów, gdyż większość badanych raczej wykazuje bierny sposób wypoczynku, a regularne podejmowanie aktywności fizycznej stanowi nie tylko profilaktykę cukrzycy, ale aż 35 różnych chorób przewlekłych.”

  1. The conclusion should be brief on your findings and discussion, and give the recommendation for policy makers and further research needed. The second paragraph in the conclusion is not too much informative as a conclusion. The generalizability of the study must be reported in the manuscript.

We have removed this fragment: ”There is a need for prevention and health promotion programs for menopausal women, with particular emphasis on people with type 2 diabetes, especially focused on the issue of excess body weight, diet, and physical activity” and we transfer this fragment from conlusion to Limitation: “A practical dimension of the presented research may be the proposal to introduce additional examinations for peri-menopausal women, based on the analysis of body composition, which will undoubtedly contribute to better planning of preventive measures in this age group”.

We are very grateful for giving us the possibility of improving our manuscript. The article has been corrected according to the Reviewers’ suggestions.

Kindest regards,

Authors

Reviewer 2 Report

Manuscript ID: ijerph-1371677

Journal: International Journal of Environmental Research and Public Health

Authors:  Małgorzata Zimny, Małgorzata Starczewska, Małgorzata Szkup, Anna Cybulska, Elżbieta Grochans

The manuscript by Małgorzata Zimny, Małgorzata Starczewska, Małgorzata Szkup, Anna Cybulska, Elżbieta Grochans „ Body composition and biological functioning in perimenopausal women with type 2 diabetes” describe the research on comparing body composition and levels of biochemical blood parameters and identify relationships between biochemical parameters and body composition of women with type 2 diabetes and healthy ones, both in perimenopausal period.

The researches carried out are very interesting. The author has conducted an in-depth comparative analysis of his results with studies conducted by other scientists. The article has been prepared reliably and, in my opinion, is suitable for publication in its current form.

further:

the author could compare the level of biochemical blood parameters with the diet and physical activity used by the study participants. The two main causes of type 2 diabetes are genetic and environmental factors, which are influenced by lifestyle. Obesity, excessive alcohol consumption and a sedentary lifestyle lead to the development of insulin resistance, which underlies the development of type 2 diabetes. Moreover, the research group consisted of 172 perimenopausal women aged 45-65 years. Were similar studies conducted, but in the pre-menopausal period (e.g. 30-45 years)? An analysis of the influence of female age would also enrich the discussion in the manuscript. Describing these aspects would make the work unique.

Author Response

Dear Sir or Madam,

We are very grateful for the review of our article titled “Body composition and biological functioning in perimenopausal women with type 2 diabetes”. We would like to thank you for all your comments and suggestions, which helped us to improve our manuscript. We are grateful for good words about the general value of the manuscript, it is very motivating for us.

The following corrections have been introduced in order to address the suggestions of the Reviewer 2 (marked in the manuscript with yellow):

  1. the research group consisted of 172 perimenopausal women aged 45-65 years. Were similar studies conducted, but in the pre-menopausal period (e.g. 30-45 years)? An analysis of the influence of female age would also enrich the discussion in the manuscript. Describing these aspects would make the work unique.

            Bardzo serdecznie dziękujemy za ten komantarz. Badania, których wyniki opierają się na analizie składu ciała nie są bardzo popularne, dlatego trudno o szerokie porównania pomiędzy grupą kobiet w okresie peri- I postmenopauzalnym. Sugestia Recenzenta skłoniła nas do zastanowiania się nad przeprowadzeniem podobnego badania w przyszłości. Niemniej udało nam się wzbogacić dyskusję o nastepujący fragment: “Warto dokonać porównania pomiędzy grupą kobiet w okresie perimenopauzalnym objętych naszym badaniem, a grupą kobiet w okresie przedmenopauzalnym. Badania przeporowadzone przez Yeung and et. Wśród premenopausal African-American and European-American women wskazały that the ratio of trunk fat to limb fat is significantly inversely correlated with sex-hormone binding globulin, which is associated with increased risk for type 2 diabetes after controlling for homeostasis model assessment–insulin resistance, estradiol, physical activity, and caloric intake. Intra-abdominal adipose tissue is an important fat depot that is associated with an abnormal metabolic profile and significantly associated with sex-hormone binding globulin independent of insulin in a female population. Despite these reports, the associations among intra-abdominal adipose tissue, insulin, and sex-hormone binding globulin in premenopausal women are unclear.”

We are very grateful for giving us the possibility of improving our manuscript. The article has been corrected according to the Reviewers’ suggestions.

Kindest regards,

Authors

Reviewer 3 Report

This manuscript is well structured but revision on the language organizing is needed. Besides, in the materials and methods part the authors introduced that two stages were conducted in this study. But results related to body composition (first stage) was presented in another manuscript. Therefore, I would like to suggest revising the methods part to fit results only presented in this manuscript or adding reference to clarify the relationship between results in this manuscript and data already published.

Author Response

Dear Sir or Madam,

We are very grateful for the review of our article titled “Body composition and biological functioning in perimenopausal women with type 2 diabetes”. We would like to thank you for all your comments and suggestions, which helped us to improve our manuscript. We are grateful for good words about the general value of the manuscript, it is very motivating for us.

The following corrections have been introduced in order to address the suggestions of the Reviewer 2 (marked in the manuscript with green):

  1. This manuscript is well structured but revision on the language organizing is needed.

Thank you for this suggestion. We asked a native speaker to read ones again our manuscript and correct mistakes.

  1. Besides, in the materials and methods part the authors introduced that two stages were conducted in this study. But results related to body composition (first stage) was presented in another manuscript. Therefore, I would like to suggest revising the methods part to fit results only presented in this manuscript or adding reference to clarify the relationship between results in this manuscript and data already published.

We have corrected Material and Methods section in this way: ‘’The method of the diagnostic survey was used to conduct the study, with the use of the questionnaire technique. After that we made an analysis of body composition with body composition analyser Jawon Medical IOI-353 (Yuseong, South Korea). In the next stage, blood was drawn in order to carry out laboratory analysis and mark biochemical parameters (lipid panel, levels of glucose, insu-lin, CRP, glycated haemoglobin).’’ More then that we added a characteristic of study and control group: “The average age of respondents was 54.9 years (study group 58.7, control group 52.3), 56.4% lived in a city with a population greater than 100 thousand. The majority had secondary (45.3%) or higher education (33.7%) (study group: the most popular was secondary education 53.9%; control group: higher education 45.2%); 76.2% were married (study group: 76,5%; control group 76%) and 75.0% professionally active (study group: 50%; study group 91.4%).”

We added an information: ‘’The research presented in this manuscript is part of a larger research project that aimed to analyze the relationship between body composition and the functioning of women with type 2 diabetes in various aspects’’ and reference to reference list.

We are very grateful for giving us the possibility of improving our manuscript. The article has been corrected according to the Reviewers’ suggestions.

Kindest regards,

Authors

Round 2

Reviewer 1 Report

Thank you very much for your updating manuscript! However, I have still some minor suggestions.

Minor comments

  1. The location of the study “West Pomeranian Voivodeship”should be included in the title of the study.
  2. The authors still missed the generalizability of the study to show how much extent represent the findings of the study to Polish Post-menopausal women.
  3. The text in line 429-432, are not related to the limitation of the study.

Author Response

Dear Sir or Madam,

We are very grateful for the review of our article titled “Body composition and biological functioning in perimenopausal women with type 2 diabetes”. We would like to thank you for all your comments and suggestions, which helped us to improve our manuscript. We are grateful for good words about the general value of the manuscript, it is very motivating for us.

The following corrections have been introduced in order to address the suggestions of the Reviewer 1 (marked in the manuscript with grey):

  1. The location of the study “West Pomeranian Voivodeship”should be included in the title of the study.

We added the word “Polish” in the title to be more precise. In our opinion, it should be sufficient, as the detailed description of the female population covered by the study is presented in the Material and Methods section. We do not want to introduce the reader into confusion, so we have completed this information in the abstract as well: “172 women aged between 45 and 65 come from the West Pomeranian Voivodeship, Poland”. Nevertheless, we believe that it is not necessary to expand the title too much with the details of the surveyed population.

  1. The authors still missed the generalizability of the study to show how much extent represent the findings of the study to Polish Post-menopausal women.

We made every effort to ensure that the group included in the study was a representative one. Unfortunately, in this study, we were able to ensure that the group was representative only at the level of the West Pomeranian Voivodeship, and not the entire Poland. In order to clarify the sampling methodology we used, we have added the following description in the Material and methods section: “Women were recruited who represented the general population of the West Pomeranian Voivodeship in north-west Poland. The size of the study sample was established on the basis of statistical data concerning the size of the 45-64-year-old female population in the West Pomeranian Voivodeship in 2017. The confidence level was set at 95%, the maximum error at 7%, and the estimated fraction size at 0.5. The total number of women qualified for the study was 172.” 

  1. The text in line 429-432, are not related to the limitation of the study.

Thank you for this comment, we deleted this sentence.

We are very grateful for giving us the possibility of improving our manuscript. The article has been corrected according to the Reviewers’ suggestions.

Kindest regards,

Authors
